# Ultra-Processed Foods and Mental Health: Where Do Eating Disorders Fit into the Puzzle?

**DOI:** 10.3390/nu16121955

**Published:** 2024-06-19

**Authors:** David A. Wiss, Erica M. LaFata

**Affiliations:** 1Department of Community Health Sciences, Fielding School of Public Health, University of California Los Angeles, 650 Young Drive South, Los Angeles, CA 90095, USA; 2Nutrition in Recovery LLC, 1902 Westwood Blvd. #201, Los Angeles, CA 90025, USA; 3Center for Weight Eating and Lifestyle Science, Drexel University, 3201 Chestnut Street, Philadelphia, PA 19104, USA; es3344@drexel.edu

**Keywords:** ultra-processed foods, food addiction, depression, eating disorders, binge eating disorder

## Abstract

Ultra-processed foods (UPFs) like pastries, packaged snacks, fast foods, and sweetened beverages have become dominant in the modern food supply and are strongly associated with numerous public health concerns. While the physical health consequences of UPF intake have been well documented (e.g., increased risks of cardiometabolic conditions), less empirical discussion has emphasized the mental health consequences of chronic UPF consumption. Notably, the unique characteristics of UPFs (e.g., artificially high levels of reinforcing ingredients) influence biological processes (e.g., dopamine signaling) in a manner that may contribute to poorer psychological functioning for some individuals. Importantly, gold-standard behavioral lifestyle interventions and treatments specifically for disordered eating do not acknowledge the direct role that UPFs may play in sensitizing reward-related neural functioning, disrupting metabolic responses, and motivating subsequent UPF cravings and intake. The lack of consideration for the influences of UPFs on mental health is particularly problematic given the growing scientific support for the addictive properties of these foods and the utility of ultra-processed food addiction (UPFA) as a novel clinical phenotype endorsed by 14–20% of individuals across international samples. The overarching aim of the present review is to summarize the science of how UPFs may affect mental health, emphasizing contributing biological mechanisms. Specifically, the authors will (1) describe how corporate-sponsored research and financial agendas have contributed to contention and debate about the role of UPFs in health; (2) define UPFs and their nutritional characteristics; (3) review observed associations between UPF intake and mental health conditions, especially with depression; (4) outline the evidence for UPFA; and (5) describe nuanced treatment considerations for comorbid UPFA and eating disorders.

## 1. Industry Involvement in the Debate on Ultra-Processed Foods and Health

A major challenge in public health is confronting the multinational billion-dollar corporations that profit from selling products such as alcohol, tobacco/nicotine, and ultra-processed foods (UPFs, e.g., soda, chips, snack cakes) known to accelerate poor health [1]. Unfortunately, public regulatory and market intervention efforts typically rely on industry self-regulation (e.g., reformulating products to barely meet school lunch standards, such as 51% whole grain [2]), an approach that has been ineffective in preventing the adverse health outcomes associated with these commodities [1].

The mainstream nutrition field has been described as “captured”, with credentialing organizations having a suspicious history of undisclosed relationships with food, pharmaceutical, and agribusiness corporations [3]. This phenomenon has been particularly pronounced in the Academy of Nutrition and Dietetics [3], the credentialing and oversight agency for registered dietitian nutritionists, largely responsible for the nutritional management of eating disorders (EDs). The Academy of Nutrition and Dietetics published public-facing rebuttals of these claims, and concerns addressing the rigor of this qualitative research have been indexed [4]. Meanwhile, the contribution of UPFs to loss-of-control (or “addiction-like”) eating remains a point of contention and division in the mainstream nutrition and ED fields, which is a major focus of this article (see Section 5 and Section 6).

To better understand the context surrounding the debate about the role of UPFs in mental health treatment, it is prudent to consider the range of “corporate scientific activities” that influence the public discourse around nutrition [5]. Such activities include (1) in-house nutrition research related to their products (with publication bias favoring findings beneficial to their products), (2) sponsoring scientific seminars and expert meetings to create consensus policies in their favor, (3) involvement in scientific standards through policy committees, (4) publishing papers in scholarly journals (often with high article processing fees), (5) funding scientific front groups (such as the International Food Information Council), and (6) delivering nutrition education programs (providing continuing education credits) [5]. Industry-sponsored messages appear to successfully manufacture doubt about the quality of the evidence regarding harms associated with their products, thereby downplaying the risk of UPFs by increasing public uncertainty [6].

Promoting UPFs (including sugar or artificially sweetened beverages) to children is particularly troublesome since they are considered a vulnerable population, and governments should be obligated to include regulations to protect their health [7], particularly during a mental health crisis. It is well known that food corporations often add essential vitamins/minerals and fiber isolates to UPFs to make on-package health claims [8], which can confuse consumers/parents about their inherent nutritional value. Most dietary guidelines rely on nutrient-based rationales that fail to apply a contemporary understanding of many of the issues with UPFs [9], which is another focus of this article (see Section 2.2).

Meanwhile, an increasing number of nutrition professionals attribute concerns about consuming UPFs to fearmongering and “diet culture”, representing “myths” about food and eating [10]. Efforts to “debunk” concerns about how UPFs interact with homeostatic hunger can be well intentioned with the important goal of ED prevention. However, such messages can simultaneously invalidate those who struggle with hedonic eating, creating a forgotten group marginalized by the mainstream nutrition narrative, potentially perpetuating shame as well as the internalization of weight stigma [11].

Furthermore, UPFs have negatively impacted environmental health in recent decades [12,13,14]. Harmful environmental exposures (e.g., hazardous wastes and other toxins, air pollutants) disproportionately impact socially disadvantaged groups [15]. Residents in low socioeconomic status areas are exposed to more UPF advertisements than their more socially advantaged counterparts [16]. Internal industry documents revealed that food companies used the tobacco company playbook to target racial/ethnic minority groups in their marketing strategies [17]. Not surprisingly, US data show that UPF consumption is higher among younger, less educated, and lower-income groups and has increased over time [18]. Thus, the conversation around UPFs can fairly be framed as a social justice issue rather than a matter of personal responsibility, adding levels of complexity to this sociological inquisition.

Considerable research has tracked changes in the global food supply since the 1980s, highlighting increases in multiple adverse cardiometabolic and mortality-related outcomes on a parallel timeline with increasing availability of UPFs [19]. However, there has been less recognition of the mental/behavioral health outcomes associated with UPFs, which is another focus of this narrative review (see Section 3). Specifically, this paper focuses on adverse biopsychological outcomes such as depression, ultra-processed food addiction (UPFA), and other forms of disordered eating, with particular emphasis on known biological mechanisms. Finally, our discussion (Section 6) calls for a more nuanced understanding of “dietary restriction” in mainstream nutrition communication, particularly for those providing recovery services across the spectrum of disordered eating.

## 2. Defining Ultra-Processed Foods and Their Nutritional Characteristics

### 2.1. NOVA Classification for Food Processing

The NOVA classification was developed by a team at the University of São Paulo, Brazil, in 2009 led by Carlos Monteiro [20]. NOVA classifies food into four categories:Unprocessed or minimally processed foods (e.g., washed/trimmed corn on the cob);Processed culinary ingredients (e.g., oils, sugars, salts, flavors);Processed foods (a combination of NOVA-1 and NOVA-2 foods, e.g., canned corn);Ultra-processed foods (containing little, if any, intact NOVA-1 foods, e.g., corn chips).

Many forms of food processing are beneficial. For example, processing increases the palatability and storage of NOVA-1 minimally processed foods (MPFs). UPFs, on the other hand, include substances not used in culinary preparations, such as flavors, colors, sweeteners, emulsifiers, other stabilizers, and preservatives. These are used to imitate the sensory qualities of MPFs, disguise the undesirable qualities of the final product, and increase their rewarding properties to maximize profit [21]. UPFs are typically manufactured through large-scale industrial processes that cannot be entirely duplicated at home.

Several authors have suggested that the NOVA classification criteria are confusing and controversial [22]. While other classification systems based on food processing have been developed, NOVA remains the most widely used worldwide [23]. One critique is that the process of classifying foods is somewhat subjective, creating discrepancies across raters and preventing scientifically robust assignment [22,23,24]. Another argument is that UPFs take on pejorative connotations, thus leading to unnecessary and unproductive negative perceptions by the public [22]. This point may be a particularly important consideration among those with restrictive EDs who are already experiencing heightened fear and distress around food.

Based on NOVA research, UPFs now dominate the food supply chain of high-income countries and are increasing in middle-income countries [25]. As countries grow richer, higher volumes and a wider variety of UPFs are sold and consumed [26]. Ultra-processed products are made to be hyper-palatable, have a long shelf-life, and be convenient for consumption anywhere and anytime. Thus, UPFs are very profitable, which permits multinational food corporations to continue investing billions in new technologies, move into new markets, and successfully lobby for their packaged branded products [27]. There is a message in the literature, predominantly in publications wherein one or more authors disclosed food industry relationships [28,29], that such foods are necessary to feed the planet, which may have some validity given the growing inability of smaller farms to compete with Big Ag. However, processing that serves to improve shelf life or add nutrients needed (e.g., fortified rice) in lower- and middle-income countries are not characterized as UPFs. UPFs are products created for palatability and profit rather than for public health initiatives.

There is also a mainstream nutrition narrative that people with any form of disordered eating should regularly eat UPFs to achieve recovery (often referred to as “all foods fit” [30]). This messaging often confuses some patients, clinicians, researchers, and the lay public, given the emerging data on the addictive properties of UPFs and the clinical presentation of UPFA. There is a lack of consensus surrounding the concept of dietary restriction, whereas some view avoidance of UPFs as a cause or consequence of EDs, and others view consumption of UPFs as a primary driver or outcome (see Section 6).

### 2.2. Nutritional Characteristics of Ultra-Processed Foods

Current evidence suggests that nearly two-thirds of the foods consumed in the US meet the NOVA-4 classification of UPFs [18,21,31]. UPFs contribute 90% of added refined sugars [21], contain higher levels of total fat from added saturated fats [32], and are inversely associated with dietary fiber and protein intake [32,33,34]. Fiber is particularly important since it is the primary fuel for intestinal microbes and is positively associated with better mental health [35].

In a Brazilian study (*n* = 32,989), the micronutrient content of UPFs was less than half observed in MPFs [36]. Most of the essential vitamins and minerals are insufficient among those consuming higher amounts of UPFs: Vitamin A, Bs (niacin, pyridoxine, folate, B12), C, D, E, and minerals copper, iron, phosphorous, potassium, magnesium, calcium, selenium, and zinc [34,36,37]. Some B vitamins (e.g., thiamine, riboflavin) are added as enrichments to ultra-processed grain products, which may explain their presence in ultra-processed diets.

An important and often overlooked component of health-promoting eating patterns is the presence of antioxidant polyphenols such as flavonoids. These compounds are easily destroyed during ultra-processing. Not surprisingly, UPF intake is associated with lower total and five out of six flavonoid class intakes [38]. There is a synergistic interaction between polyphenols and the fibers abundant in NOVA-1 plant foods that are modulated by gut microbiota [39]. It has been hypothesized that gut microbiota may be a missing link in our current understanding of ED etiology [40].

Thus, low intakes of fiber and polyphenols in combination with high intakes of substances found in UPFs may lead to elevations in inflammatory markers. For example, multiple studies have shown that consumption of UPFs is associated with elevated C-reactive protein (CRP), interleukins (e.g., IL-6, IL-8), and tumor necrosis factor-alpha (TNF-⍺) [41,42,43]. Elevated inflammatory markers in the context of mental health (i.e., depression) are significant [44] and will be discussed below. One mechanism is the microbial encroachment to the gastrointestinal wall (leading to intestinal permeability) and the subsequent cascade of inflammatory processes throughout the periphery [42].

UPFs are generally not consumed in isolation; however, meta-analysis suggests that increased UPF intake is associated with worse diet quality (e.g., fewer fruits and vegetables) [32]. More research is needed to assess the nutritional impact of UPFs among those who consume relatively higher amounts of fibers, polyphenols, omega-3 fatty acids, and protein from whole food sources, as there is likely a threshold (e.g., percentage of UPFs in the context of the total diet) that confers health risk. Some UPFs, such as plant-based meat alternatives and protein bars, are marketed as filling nutritional gaps among those with dietary restrictions for health reasons (e.g., allergies) or other preferences. This argument appears to be part of the “planting doubt” playbook, which aims to exonerate manufacturers from other public health concerns about other UPFs (e.g., chips, pastries) more closely associated with addiction-like eating and binge-type EDs [45].

A detailed exploration of the biological mechanisms linking UPFs to health outcomes is outside the current scope; please see recent reviews [46,47,48]. To synthesize and summarize their findings, UPFs negatively impact nutritional status through the following:Displacement (consuming UPFs in lieu of MPFs);Poor nutritional profile (low micronutrient/phytonutrient density of foods);Neo-formed compounds from high-heat processes (e.g., acrylamide, advanced glycation end products, industrial trans-fats, furans);Inflammatory responses to industrial food additives (e.g., emulsifiers, preservatives, thickeners, artificial sweeteners, flavoring agents, coloring agents);Contaminants from packaging materials (e.g., bisphenols, phthalates);Higher glycemic load reducing satiety signaling (e.g., refined grains and sugars);Reduced fiber volume (leading to faster eating and decreased satiety);Gut microbiota dysbiosis (associated with the loss of beneficial and overgrowth of opportunistic microbes, reducing short-chain fatty acid production in the colon);Increased intestinal permeability (leading to peripheral inflammation);Overconsumption (addiction-like eating) (see Section 4).

## 3. Associations between Ultra-Processed Foods and Mental Health

The most convincing evidence linking UPFs to adverse mental health outcomes is found in patients with depression and depressive symptoms. Some studies have found associations between UPFs and anxiety [49], but the weight of the evidence suggests a dose–response risk for depression but not anxiety [50]. Large prospective studies have documented associations between increased UPF intake and higher incident depression after extensive adjustment for dietary patterns correlated to UPFs (e.g., physical activity, sleeping hours, television viewing) and other potential confounders (e.g., marital and employment status) [51,52]. These dose–response findings often emerge in adolescence, even after adjusting for factors such as self-perceived body image and bullying victimization [53,54]. Higher UPF consumption also correlates with elevated internalizing symptoms such as boredom, crying, fear, worry, loneliness, unhappiness, poor sleep, and sadness [55]. While UPF intake has been associated with ADHD among children [56,57], research is needed to examine this association among adults.

Among young adults (*n* = 1270), factors such as being unmarried, smoking, excess alcohol intake, negative health perception, and high perceived levels of stress levels correlate with higher UPF consumption [58]. According to a recent meta-analysis, for every 10% increase in UPF consumption relative to daily calorie intake, there was an 11% higher risk of depression in adults [50]. To date, seven randomized controlled trials have shown that depression levels significantly decrease after counseling interventions that encourage the intake of MPFs while reducing UPFs [59]. In their systematic review, studies designed specifically to induce weight loss were excluded because body weight positively correlates with depressive symptoms [60]. To date, the Mediterranean diet has the most consistent association with decreased depression risk [61,62].

Among an ethnically diverse sample of adults (*n* = 10,775) ages 35–74, a higher daily percentage of UPFs was associated with cognitive decline [63]. Several longitudinal studies have linked high UPF consumption to dementia [64]. Substituting MPFs for UPFs has been associated with a lower dementia risk [65]. Clearly, UPFs impact cognitive health over the life course. To further exemplify, pregnant women who consumed the most UPFs during their third trimester had children who later developed lower verbal scores on the McCarthy Scales of Children’s Abilities [66].

Several authors have reviewed the biological mechanisms of depression [67] and how they may be linked to diet [68]. While an extensive review of these mechanisms is outside of this paper’s scope, additional depression-specific mechanisms can augment the list above:Nutrient deficiencies (B vitamins, vitamin D, omega-3, magnesium, zinc, selenium, iron) [69,70];Oxidative stress (which can lead to inflammation) [68,71,72];Blood–brain-barrier permeability (associated with intestinal permeability) [71,73];Hypothalamic–pituitary–adrenal (HPA) axis (responsible for stress response via cortisol) [68,70];Neuroendocrine hormones (e.g., leptin and insulin resistance) [74,75,76];Endocrine disruption (from neo-formed compounds, food additives, and contaminants) [71,77];Mitochondrial dysfunction (responsible for energy production) [68,78];Brain function and volume (functional connectivity related to eating modulation) [72,79];Brain integrity (distorted fatty acid composition of brain membrane phospholipids) [69,73];Brain-derived neurotrophic factors (BDNF) (related to synaptic plasticity) [80,81];Serotonin neurotransmission (tryptophan diverted to kynurenine rather than serotonin) [68,82];Dopamine overactivation (related to addiction-like eating) (see Section 4).

Many of the mechanisms listed above reflect alterations at multiple levels of the gut–brain axis [83]. A daily multivitamin/mineral will not address the list. It is increasingly recognized that the gut microbiota is also involved in shifts from homeostatic to hedonic mechanisms of food intake. The next section discusses hedonic eating through research on food addiction, which will be referred to as ultra-processed food addiction (UPFA), adding the specificity that UPFs (most, but not all) are uniquely related to this concept [84,85].

## 4. Evidence for Ultra-Processed Food Addiction

The emerging field of food addiction is not without criticism and debate [86,87,88,89], and the condition is not a formally recognized diagnosis. While the Yale Food Addiction Scale (now YFAS 2.0) [90,91] asks about symptoms and behaviors related to foods that can be categorized as UPFs (e.g., sweets, packaged snacks, sweetened beverages), the NOVA classification system has substantiated a proposed rename of “food addiction” to the more specific term UPFA. Notably, over a decade of research has demonstrated that most UPFs elicit addiction-like biological and behavioral responses, whereas MPFs do not [92]. Biological responses include striatal activation (i.e., dopamine projection), whereas a more frequent consumption pattern reduces reward responsivity in this region [93]. Such changes parallel the phenomenon of tolerance [94], which is associated with UPF-seeking despite risk and negative consequences [95]. While there is no single biomarker for addictions, neuroimaging research on functional connectivity [96] that corresponds with behaviors associated with decreased executive functioning (e.g., impulsivity) around salient stimuli (i.e., UPFs) will strengthen the UPFA construct. The role of early life adversity is emerging as an important predictor of altered reward network connectivity among those with UPFA [97].

Some estimates suggest that the prevalence of UPFA, as assessed by the YFAS 2.0, is as high as 20% in the general population [98]. Other estimates suggest it is closer to 14% among adults [99] and approximately 12% among children [100], a level of addiction unprecedented among youth. Notwithstanding, factors such as consumer perspectives and stigma (internalized and externalized) related to addiction-like eating are somewhat sparse in the literature and require further investigation before this construct might reach wider acceptance.

A randomized controlled trial of weight-stable adults (*n* = 20) demonstrated that diets comprising UPFs increased caloric intake by over 500 kcal/day [101]. Participants gained weight on the UPF diet and lost weight eating MPFs. One mechanism may be reduced secretion of the hunger hormone ghrelin and increased levels of the satiety hormone PYY with the unprocessed diet. However, ghrelin is not associated with UPFA, while leptin (which regulates energy balance by suppressing hunger) is [102]. While increased hunger and decreased satiety hormones contribute to the likelihood of hedonic eating, dopaminergic sensitization and increased connectivity within the brain’s reward network are more likely to explain the reward-driven motivation for UPFs [103,104]. Another proposed contributing mechanism includes changes in gut microbiota [104]. A systems biology approach that integrates the interplay of these various mechanisms has recently been proposed [83].

Cross-sectional data have linked UPFA to higher consumption of NOVA-4 foods, particularly sources with added sugar [105,106]. For each additional UPFA symptom reported on the YFAS 2.0, the percentage of energy from UPFs was higher, and the percentage of energy from MPFs was lower [107]. Increased intake of UPFs would be expected among individuals exhibiting an addiction-like response to these foods. Because UPFs can directly cause biological changes (e.g., reward sensitization) among susceptible individuals, they pose unique challenges for dietary change/improvement [108], particularly when homeostatic dietary interventions (i.e., portion control) fail to identify the contribution of UPFs to hedonic processes. One explanation related to personal risk factors might be that individuals with UPFA have consistently exhibited higher levels of impulsivity and negative urgency [109]. In a food environment laden with convenient access to inexpensive UPFs, it is not surprising that nutrition habits are challenging to change, particularly in the face of various life stressors and ongoing psychosocial adversity [110,111].

## 5. Ultra-Processed Foods and Eating Disorders

It has been suggested that UPFs are a “blind spot” in ED research and treatment [112]. The mindset in contemporary ED treatment is that “all foods fit” [30], which was designed to combat diet culture among those with (or at risk of) restrictive disorders of eating. One origin of this philosophy stems from Fairburn’s Transdiagnostic Theory, which proposes that common mechanisms are involved across the spectrum of disordered eating [113]. Another origin stems from a position paper from the Academy of Nutrition and Dietetics known as the “Total Diet Approach to Healthy Eating” [114], which states that “all foods can fit within this pattern if consumed in moderation with appropriate portion size” which appears to be a sensible, yet industry-friendly message (not “demonizing” any specific foods or ingredients, putting the onus of inhibition on the individual).

While ED treatment teams generally include a psychiatrist and therapist, registered dietitian nutritionists are ideally responsible for directing eating patterns during care. It is unsurprising that food philosophies in most treatment centers reflect a position of inclusivity around UPFs, given that many presentations of restrictive EDs (e.g., anorexia nervosa) attempt to diet in harmful ways and eliminate many foods unnecessarily. Thus, adopting an “all foods fit” philosophy may be critical for many individuals seeking treatment to ameliorate these behavioral and psychological extremes. This inclusive approach aligns with social justice issues related to weight stigma within the anti-diet movement and warrants prioritization in many clinical interventions. Weight-inclusive counseling strategies can be effective but should always be personalized to individual needs [115] rather than implemented at scale. We argue that integrating the evidence linking UPFs to mental health reviewed herein does not counter the premise of Health at Every Size^®^ as a social justice movement but rather examines the issue at the structural level.

For other individuals with EDs, particularly binge-type EDs, and more complex, comorbid symptomatology, such as concurrent UPFA, substance use disorder, depression, and/or gastrointestinal conditions, a more nuanced nutritional approach may be needed [116]. These authors suggest that an assessment of dietary restraint in conjunction with early life adversity and post-traumatic stress disorder symptoms (as well as the temporal sequence of disorder onset) may be informative when assessing the appropriateness of addressing addiction-like eating patterns of UPFs during efforts to provide treatment and care.

Based on extensive experience from the field, very few treatment centers acknowledge the role of UPFA during ED treatment; some insist that the condition does not exist, and others believe it to be a harmful concept (by stimulating unwarranted fear and rules). Little attention is given to food quality, and food avoidance is actively discouraged [112]. Social and individual psychological factors (such as media influences and weight stigma) are considered more important than biologically based behavioral responses to food. This position ignores the direct contributions of UPFs in driving overconsumption among individuals exhibiting UPFA. Meanwhile, the comorbidity of UPFA and EDs has been associated with worse clinical conditions and symptoms [117,118], which may explain higher treatment dropouts among those with addiction-like eating [119].

Many in the ED field have accepted the calories-in, calories-out hypothesis without any criticism [112]. While increasing caloric intake may be the first treatment priority for some underweight patients, the calorie model does not adequately capture other key biological mechanisms of food and eating behavior. According to one study of ED patients in the UK (*n* = 73), those with anorexia nervosa consumed 55% UPFs (slightly less than the population average daily UPF consumption), while this estimate increased to approximately 70% (close to the population average) among those with bulimia nervosa and binge eating disorder [45]. However, in this study, 100% of foods consumed in binge episodes were UPFs, strongly implicating UPFs in binge eating behavior. It is important to note that some patients with EDs gravitate toward diet products (believing them to be healthier options), which are commonly UPFs (e.g., diet beverages), which may be less commonly implicated as addictive. Individuals with EDs might also choose UPFs because they usually have calorie counts on them (whereas MPFs often do not). Another explanation that may be particularly likely (given the high degree of comorbidity between binge-type EDs and UPFA) is that most UPFs are more reinforcing and hedonically salient due to their ability to alter dopaminergic processes and directly motivate consumption despite negative consequences (as seen in substance use disorders [120]). Observational research has found that UPF consumption is positively associated with bulimia nervosa and binge eating disorder but not with purely restrictive EDs [121].

Higher intake of UPFs has been identified as one precursor to binge eating in children and adolescents [122]. Other drivers cited in this study include genetics, parenting style (affecting attachment pattern and child temperament), the gut microbiome, and lifestyle factors (e.g., television viewing). It is widely accepted in the ED field that all binge eating stems from some form of dietary restraint (commonly referred to as “restriction”), often generated by body dissatisfaction [123]. The available evidence does not support this absolute assertion [124]. On the contrary, longitudinal research using an adolescent sample found that UPFA predicted dietary restraint, but dietary restraint did not predict UPFA [125]. This is not to suggest that UPFA is synonymous with binge eating; although, their overlap is significant [126]. Addiction-like eating may drive the initial dietary restraint, which may reflect an attempt to protect oneself from the overstimulating properties of UPFs, which may then, in turn, be overridden by the anticipated reinforcement from UPFs and result in binge behaviors [123]. Not all binge eating episodes are directly linked to restriction [124]; the abundance of UPFs in our food environment may result in cue-induced craving and promote binge-like consumption patterns (i.e., the blind spot).

Consequentially, the upstream factor of UPFA is often ignored in ED treatment because it challenges the “all foods fit” philosophy that affirms dietary restraint is the key causal driver of ED symptomology. Furthermore, it is often unclear how to best manage UPFA in a food environment inundated with easy access to these convenient foods. While “abstinence” is common in most addiction treatments, flexible (rather than based on rigid rules) harm reduction models have been proposed [84]. There is evidence for the efficacy of harm reduction models for other socially acceptable substances like alcohol [127].

The most confusing research on UPFA and EDs is with respect to anorexia nervosa. Approximately 44% of individuals with anorexia nervosa meet the criteria for UPFA (with 48% for bulimia nervosa and 55% for binge eating disorder) [98]. It has been suggested that individuals with the restrictive type of anorexia nervosa restrain their eating to control exposure to certain UPF stimuli (e.g., chocolate) associated with hypersensitive neural responses [128]. On the other hand, increased neural responses could also result from excessive restraint (starvation response). Some research has suggested two distinct phenotypes of anorexia nervosa: one with restrictive features secondary to addiction-like eating and one “originally restrictive” without clear addiction-like mechanisms [129]. Other research has found that individuals with restrictive anorexia nervosa who screen positive for UPFA share more similarities (general psychopathology and personality traits) with the binge-purge type than those with the restrictive type without UPFA [130]. Thus, the presence of UPFA might be an indicator of crossover from restriction to binge-purge. Most likely, individuals with anorexia nervosa exhibit high levels of UPFA because the most common tool for assessing UPFA, the YFAS 2.0 questionnaire, is subjective and based on self-reported symptoms that may be perceived as addiction-like (e.g., an exaggerated perception of unsuccessful efforts to cut down, loss of control, etc.) [84]. Notwithstanding, given that patients with UPFA exhibit different profiles than individuals without, personalized nutritional treatments that go beyond “all foods fit” may be warranted [116]. For example, recent case reports have described remissions from anorexia nervosa with the ketogenic diet combined with ketamine [131,132,133,134].

## 6. Discussion

This article provides a backdrop of the food industry’s strategies in promoting and defending UPFs despite emerging evidence of their mental health risks. One method of planting doubt about the harms of these foods has been through the prevailing narrative that avoiding UPFs is a pathological form of restriction associated with the development and maintenance of ED symptoms. While this may be the case in some presentations, we argue that it is not true for all forms of disordered eating, particularly when there is comorbid depression and UPFA.

We reviewed biological mechanisms that contribute to associations between increased UPF intake and greater incidence rates of mental health conditions, especially depression, UPFA, and disordered eating. A mechanistic understanding of the deleterious effects of UPF is underway and may provide additional insight into the nutritional management of other mental health conditions, such as post-traumatic stress disorder. Many of these pathways explaining the connection between UPFs and depression also overlap with the biological embedding of adversity, reviewed elsewhere [135]. In contrast, research on the role of UPFs in EDs has only begun to emerge; although, the available evidence documenting the cooccurrence of UPFA and EDs is robust.

For many, it is taboo to discuss the qualities of food (e.g., organic vs. conventional, degree of processing) in any ED context since many individuals are sensitive to this kind of messaging. Clinicians fear that it can trigger symptoms of orthorexia nervosa, which is an unhealthy preoccupation with the qualities of food [136] and perpetuate further pathological restriction. Meanwhile, many patients who present for ED treatment inquire about addiction-like eating [137]. According to this Australian study (*n* = 175), 60% of professionals reported interest in receiving training in addictive eating. An unmet need in clinical practice clearly exists for both patients and practitioners.

It is critical to acknowledge that dietary restraint and restriction are key drivers of many forms of disordered eating [116]. Meanwhile, more nuance is needed to define dietary restraint, as not all forms are pathological [138]. For example, if an ED patient has unresolved depression and would like to experiment with a Mediterranean diet (high in fiber, polyphenols, and anti-inflammatory fats) to improve their mental health [139], would that be considered pathological? Given that more than half of ED patients have a mood disorder [140], it appears to be a question worth posing. What if they have autoimmune issues such as Hashimoto thyroiditis or multiple sclerosis or have gastrointestinal issues like inflammatory bowel disease or irritable bowel syndrome? Recent evidence suggests that reducing UPFs may improve all of these conditions [141,142,143].

More relevant to the evidence reviewed herein, what about the individuals with binge-type EDs who clearly meet the criteria for UPFA? Do all foods fit? Probably not, as continued consumption of UPFs may directly maintain binge eating patterns among these individuals. In these cases, a harm reduction approach could be considered to nuance the pursuit of abstinence from specific UPFs that consistently trigger binge eating while pursuing an ongoing practice of moderation with other UPFs that do not lead to problematic overeating. Efforts to address the internalization of weight stigma, body shame, and shame-based eating can accompany this approach and are not counteracted by the science of UPFA. Similar psychological treatments used for EDs can support UPFA, but the food plans might differ. For severe cases of UPFA, a rigid rule-based approach removing all guesswork from daily food choices may be the only path to positive mental health. However, when addiction recovery is conflated with (or measured by) weight loss, it can easily be perceived as an excessive (i.e., pathological) form of restraint that most ED professionals would aim to treat.

Interestingly, ED clinicians and researchers have been shown to have a decreased ability to seek alternative solutions compared to general mental health clinicians and researchers [144]. Professionals working with EDs should become more flexible with food philosophies to treat a wider range of clinical presentations. This can be difficult when treatment is scaled, and food service poses operational and financial challenges. The cognitive load of clinicians cross-identifying with seemingly contradictory approaches can also create barriers to implementation, especially if education is provided in group settings or if there is a personal bias based on clinician recovery and an allegiance to a particular group. To assist with this, Table 1 details forms of dietary restraint in the context of disordered eating. It identifies different types of nutrient restriction, disordered characteristics often associated with them, and appropriate nuance given the emerging science of personalized medicine at the intersection of nutrition and mental health.

## 7. Strengths and Limitations

A strength of this review is that it merges expert experience from patient care in the fields of ED and mental health with the latest available evidence. The paper summarizes recent research to offer clinical considerations for professionals involved in the treatment of disordered eating. A limitation is the lack of available data on interventions for the concurrence of EDs with depression and UPFA. No intervention trials have focused on food quality in treating EDs; thus, this work may inform future efforts to investigate the role of UPFs in ED pathology and recovery.

## 8. Conclusions

According to Dr. Monteiro, UPFs are troublesome from social, cultural, economic, political, and environmental points of view [27]. Emerging data exist that UPFs are linked to compromised mental health worldwide. These foods are produced for corporate profits at the expense of public health. The nutritional inferiority of these foods is clear when compared to MPFs. Meanwhile, the changing human brain is more susceptible to convenience and the overvaluation of neurochemical rewards (salience), especially when distressed. Thus, many “choose” to eat UPFs even after understanding their implications. The increasing domination of industrial food processing has replaced traditional eating patterns, and worldwide trends in EDs closely parallel this transition [112]. Maybe it is “about the food” [148].

Overconsumption of UPFs may be a cause and consequence of symptoms of depression, UPFA, and disordered eating. An ED diagnosis should not automatically warrant an “all foods fit” treatment approach that precludes the direct contributions of UPFs to exacerbated eating pathology and comorbid mental/physical health concerns for some individuals. According to Dr. Ayton, eliminating UPFs (including in ED treatment) and diet foods (e.g., diet soda) could help restore normal satiety mechanisms [112]. This may not be culturally acceptable given the ubiquitous presence of these foods and the large percentage of the lay public who enjoy them without physical consequences or psychological distress. Until we can mitigate the corporate capture of the nutrition field and regulate big industries, we are left fighting amongst ourselves about divergent food philosophies (i.e., whether to address the role of UPFs in treatment and recovery) that appear to embody equally important and overlapping social justice implications. It seems wiser for treatment providers to join forces than to oppose each other.

If “food neutrality” is a goal, some may achieve it through strategic inclusion of all UPFs (i.e., the “all foods fit” approach), and others may achieve it through strategic avoidance of specific, triggering UPFs (e.g., harm reduction). We desperately need more dialectics in dietetics. We teach our patients to be less rigid and not think in black-and-white terms, but many professionals have adopted this cognitive distortion regarding food philosophy. The tension is palpable. As a field, we can do better. Our patients deserve practitioners prioritizing individualized intervention planning over dogmatic adherence to a singular (antiquated) treatment philosophy.

## Figures and Tables

**Table 1 nutrients-16-01955-t001:** The nuance of dietary restraint (“restriction”) in the context of disordered eating.

Type of Restriction	Disordered Characteristics	Nuance
Calories	Counting and tracking Only eating food with known calories Ignoring all other nutritional details of food	Reducing calories (through improved food choices rather than counting) can be a part of recovery for some people whose disorder is centered on excess, impulsivity, and mindless eating.
Macronutrients	Attributing a macronutrient to undesirable body shape Tendency toward dietary extremes to manage perceived crisis of living in a body that does not feel like home (to lose weight)	Macronutrients also have important implications for brain health. In one’s quest to achieve mental wellness, it is possible that lower protein approaches (e.g., plant-based [145]) or lower carb (e.g., ketogenic [146]) can stabilize the brain and thereby support mental health.
Food Groups	Assigning negative valence to an entire food group based on beliefs about the entire category or due to calories/macros Diet defined by exclusion for a socially acceptable form of restraint	Not everyone needs to eat all food groups to achieve adequate nutrition [147]. Some exclusions (e.g., dairy) can be culturally based, and others (e.g., animal protein) can be ethically based or simply due to preference. Alternative “dairy” and plant-based “meats” can work well for some but not all.
Food Quality (Organic)	Symptoms of orthorexia nervosa characterized by heightened fear of the food supply chain Prioritization of perceived physical health at the expense of mental health	Interest in the agricultural practices of food is pathological only if it leads to other forms of restriction (e.g., refusing to eat non-organic food when hungry). Many people find organic food tastes and digests better which is not necessarily disordered.
Food Processing (NOVA)	Assigning negative valence to food based on its degree of processing and/or packaging Avoidance of ultra-processed foods for weight loss purposes or based on magnified fears	Some people benefit from eating less ultra-processed foods due to depressive symptoms, addiction-like processes, auto-immune issues, or gastrointestinal complaints. Others choose to consume less of these foods for spiritual or environmental reasons, which is not necessarily disordered.

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
