# Peer review of "Ultra-Processed Foods and Mental Health: Where Do Eating Disorders Fit into the Puzzle?"

_nutrients, 2024, doi:10.3390/nu16121955_

Round 1
Reviewer 1 Report
Comments and Suggestions for Authors
Thank you for the opportunity to review this paper on a topical issue. I have some concerns particularly regarding that the background and conclusion do not summarise your aims of the paper with little to no focus on UPFs, mental health and EDs. This makes it difficult to read and follow what the key point of this paper is as the background suggests the focus is on the food industry, with little mention of UPF, health and mental health. Therfore, I would recommend reworking this paper so that it more clearly aligns with what is presented in the abstract as the aims.
Abstract - I am not sure that point 4 of what this review is outlining is the right terminology, would it be better to say describe treatment considerations or something along those lines?
Background - majority of the background focuses on the issues with the food industry and nutrition research and doesn't really give any context to why you are doing this review or match your 4 key aims of the review. Potentially, you could focus more on why you are focusing on UPF, mental health and eating disorders as this is only briefly mentioned in the background and is not a clear starting point for your review.
Perhaps you could more clearly outline your aims at the end of the background so they cover all the same things as what is listed in the abstract.
Do you have a reference to support this statement in the NOVA section "There is also a dietitian-endorsed narrative that all people with all forms of disordered eating need to regularly eat UPFs to achieve recovery (valid in some, but not all, cases)." Additionally, I am not sure if this entire paragraph belongs in the section outlining the NOVA classification system. Perhaps this paragraph could be removed or moved elsewhere where it makes more sense with what you are discussing.
For the list on page 4 and 5 - it would be better to put each individual reference supporting the statement after each point rather than grouping them together prior to the list.
On page 7 do you have a reference to support this statement ". Other drivers include genetics, parenting style (affecting attachment pattern and child temperament), the gut microbiome, and lifestyle factors (e.g., television viewing)"
Should point 8 be "Strengths and limitations"?
Conclusion - overall this conclusion doesn't really summarise the findings of your 4 aims in your abstract or your findings. Instead, similar to your background section it appears to mostly be about your opinions on food industry. While it is important to consider, it comes across as this is the key focus of your paper not the relationship between UPF and mental health and EDs.
References 67-69 are all the same reference
Reviewer 2 Report
Comments and Suggestions for Authors
The review provides an interesting summary of UPF and mental health from, citing examples of the authors own previous work as well as from existing scientific literature. The review highlights important considerations for eating disorder management. Some suggested modifications are below, that relate mainly to places in the review that I believe could be a bit more balanced and the language tempered. There are also arguments throughout where references could be added to support the claims made. Overall, this is a timely and needed review that will implications for understanding and treating eating disorders.
1. Background
Please provide references for the following sentence: “A major challenge in public health is confronting the multinational billion-dollar corporations that profit from selling products such as alcohol, tobacco/nicotine, and ultra-processed foods (UPFs, e.g., soda, chips, snack cakes) known to accelerate poor health.”
To substantiate this sentence “This phenomenon has been particularly pronounced in the Academy of Nutrition and Dietetics…” Carriedo et al. could be cited here also. To provide balance, it should also be noted that the Academy of Nutrition and Dietetics have made public statements to refute this claim.
2. NOVA Classification for Food Processing
Although NOVA seems to be the most widely used system, for the benefit of readers acknowledgement that other classification systems exist is needed. Some discussion on the issues around definitions and terminology would also show a critical appreciation of the relevant issues.
It could be noted in this section, to balance the arguments, that some low- and middle-income countries require processing for food fortification to prevent nutritional deficiencies.
“There is also a dietitian-endorsed narrative that all people with all forms of disordered eating need to regularly eat UPFs to achieve recovery” Please provide the evidence that supports this point.
3. Ultra-Processed Foods and Nutritional Status
What is not acknowledged here, is that people do not consume UPFs in isolation. Often people have high intakes of UPF, while also consuming high intakes of core foods e.g. fruits and vegetables. It should also be mentioned somewhere that some UPFs are necessary for people with special eating restrictions, such as allergies etc.
The concept of ‘leaky gut’ is not widely recognised as a medical condition by healthcare professionals.
4. Ultra-Processed Food and Mental Health
Given the aims of the review, and the authors state “Specifically, this paper focuses on adverse outcomes such as depression…” I find it interesting that dementia has been included here. While people with dementia may experience mental ill-health, dementia as such is not a mental illness.
While an extensive review of “the biological mechanisms of depression and how they may be linked to diet” may be outside the scope of the review, there are many systematic reviews that demonstrate the association between diet and depression that could be cited in this section to provide a higher level of evidence.
5. Ultra-Processed Food Addiction
While the authors acknowledge there is ongoing debate surrounding food addiction, there is a lack of acknowledgement that food addiction (or UPFA) is not currently recognised as a diagnosable condition or eating disorder within the DSM or WHO.
Given there is currently no accepted definition/terminology, I feel the authors are nudging the food addiction field toward a particular framing which is not yet accepted in the scientific literature. The following sentence needs to be tempered “…the NOVA classification system has substantiated refinement of the vague “food addiction” term to the more appropriate UPFA.” Although the authors present a compelling case for UPFA (defined by NOVA), at present it is more widely accepted as ‘food addiction’ given the specific chemicals in foods that are addictive have yet to be identified. More research is needed as there are further factors that also need to be considered here, such as consumer perspectives, health perspectives and stigma.
I find the following to be rather general, “…over a decade of research has demonstrated that UPFs elicit addiction-like biological and behavioral responses, whereas MPFs do not”. I suggest this point be expanded provide more detail (e.g. human or animal research) including examples of biological and behavioral responses.
It is briefly mentioned in the discussion, but I suggest acknowledge in this section that further research is needed to determine exactly how UPFs trigger an addictive response to be able to understand causal pathways.
6. Ultra-Processed Food and Eating Disorders
I feel this section needs to reflect the broader eating disorder field, as there are many health professionals, not just dietitians, involved in treatment of eating disorders (e.g. psychologists, psychiatrists) depending on the severity.
The second paragraph could also mention ‘Health at Every Size’ as an approach that “aligns with social justice issues related to weight stigma”. See recent systematic review by Clarke et al. (https://doi.org/10.1111/1747-0080.12869) that provides support for personalising treatment to individual needs.
“Very few treatment centers acknowledge the role of UPFA during ED treatment; some insist that the condition does not exist, and others believe it to be a harmful concept (by stimulating unwarranted fear and rules).” Please provide the evidence that supports this point.
Please specify in which country the study population were from: “According to one study of ED patients, those with anorexia nervosa… slightly less than the population average daily UPF consumption…”
Please provide references for the following sentence: “Other drivers include genetics, parenting style (affecting attachment pattern and child temperament), the gut microbiome, and lifestyle factors (e.g., television viewing).”
Minor comments
Where specific studies or systematic reviews are discussed (e.g. “In a Brazilian study…”; “Large prospective studies…”) it would be beneficial for the evaluation of findings if study sample sizes were included (n=).
Weblink for the first reference appears not to be working.
Reviewer 3 Report
Comments and Suggestions for Authors
Thank you for the opportunity to review this interesting narrative review exploring UPF’s and mental health. This is a well written paper that covers an important topic, however, there are some minor modifications required.
I understand the intention of the piece to stimulate debate, however, the language around dietitians, organisations, and ED dietitians implies all dietitians are complicit in recommending UPFs. It would be useful to revisit the language in these sections as well as provide a more balanced view on the position of many dietitians regarding UPF, including work being done in the food marketing space, for example: Home Page - Food For Health Alliance. Moreover, while the focus is on dietitians as the ‘gatekeepers’, in certain ED treatments such as FBT, the family are responsible for refeeding the child, with support from a therapist. The role of the family could be explored in the section in eating disorders.
Although based on current literature I can understand why the focus between UPF and mental health is on depression, it would be useful to also highlight other mental health conditions in further detail (e.g. PTSD which is discussed later in the paper, ADHD, anxiety), gaps and future directions in UPF in these mental health areas. Alternatively, this subsection heading may be altered to focus on depression, rather than mental health generally.
Table 1: Further descriptive information is needed in the Table 1 description. For example, what is meant by “nuance”, “hierarchy of dietary restraint”. Rephrasing “sanity over vanity” would be useful, as this contributes to potential stigma that eating disorders are a choice based on appearance. Some references would be helpful for table 1, for example, meeting nutrition requirements when excluding food groups, ketogenic diet etc to better understand the context of these statements.
